# The Involvement of Antioxidant Enzyme System, Nitrogen Metabolism and Osmoregulatory Substances in Alleviating Salt Stress in Inbred Maize Lines and Hormone Regulation Mechanisms

**DOI:** 10.3390/plants11121547

**Published:** 2022-06-10

**Authors:** Mingquan Wang, Shichen Gong, Lixin Fu, Guanghui Hu, Guoliang Li, Shaoxin Hu, Jianfei Yang

**Affiliations:** Maize Research Institute, Heilongjiang Academy of Agricultural Sciences, Harbin 150086, China; ymswmq@haas.cn (M.W.); gongsc@126.com (S.G.); flx0802@163.com (L.F.); liguoliang111@126.com (G.L.); xinxin_future@126.com (S.H.); jfyang07@126.com (J.Y.)

**Keywords:** inbred maize lines, salt stress, antioxidant enzyme system, nitrogen metabolism, osmoregulatory substances, hormone regulation mechanism

## Abstract

Salt stress inhibited the growth of maize. B46 and NC236 were chosen as materials and NaCl concentrations (0, 55, 110, 165, and 220 mmol L^−1^) were set. We found the activities of SOD, POD, CAT, APX, GR, MDHAR, and DHAR decreased under NaCl stress. Compared with NC236, the contents of AsA and GSH, AsA/DHA and GSH/GSSG of B46 decreased. The content of O_2_^−^, H_2_O_2_, MDA, and EL of B46 increased. The contents of NO_3_^−^ and NO_2_^−^ decreased, while the content of NH_4_^+^ increased under high NaCl concentration. The activities of NR and NiR decreased, while the activities of GS and GOGAT increased first and then decreased. For B46 and NC236, the maximum of NADH-GDH and NAD-GDH appeared at 165 and 110 mmol L^−1^ NaCl concentration, respectively. Compared with B46, and the GOT and GPT activities of NC236 increased first and then decreased. With the increase of NaCl concentration, the contents of proline, soluble protein, and soluble sugar were increased. The Na^+^ content of B46 and NC236 increased, and the K^+^ content and K^+^/Na^+^ decreased. Compared with NC236, B46 had higher IAA content in leaf, higher Z + ZR content in leaf and root, and lower ABA content in leaf and root.

## 1. Introduction

Soil salinization has become a global problem and the biggest obstacle to agricultural production [1]. China’s saline-alkali land covers an area of 100 million hectares. SongNen Plain, located in northeast China, is one of the most widely distributed areas of saline-alkali land globally. Soil salinization inhibits crops’ physiological and biochemical processes and eventually leads to declining in grain yield and quality [2]. Cultivating salt-tolerant plants and improving their salt tolerance is an effective biological measure to alleviate the impact of saline alkali land on plants. At the same time, it can also produce better ecological and economic benefits and promote the sustainable development of agriculture. Excessive soil salt accumulation reduces the osmotic potential of soil solution severely [3,4,5]. The balance of root water absorption was blocked between water absorption and transpiration loss [6]. Salt stress can directly lead to stomatal closure and inhibit the carbon assimilation process of photosynthesis [7]. ROS accumulation destroys biological macromolecules in the cytoplasm and interferes with normal metabolism in plants [8]. Superoxide anion (O_2_^−^) in leaves of plant seedlings production rate and hydrogen peroxide (H_2_O_2_) content increase with the increase of NaCl concentration [9]. O_2_^−^ in plant seedlings under salt stress production rate and H_2_O_2_ content increased first and then decreased with NaCl treatment time [10]. In the ascorbic acid glutathione (AsA-GSH) cycle, ascorbic acid peroxidase (APX) uses reduced ascorbic acid (AsA) as an electron donor to catalyze H_2_O_2_ to non-toxic H_2_O and produce two molecules of monodehydroascorbic acid (MDHA) at the same time. Subsequently, ascorbic acid reductase (MDHAR) uses NADPH as an electron donor to reduce MDHA to AsA. MDHA was catalyzed by dehydroascorbic acid reductase (DHAR) to dehydroascorbic acid (DHA), and this physiological reaction takes reduced glutathione (GSH) as the substrate and produces oxidized glutathione (GSSG) [11]. GSSG will be reduced to GSH again by the catalysis of glutathione reductase (GR) for a new cycle [12].

At the same time, the physiological process of plant salt tolerance is related to nitrogen metabolism [13]. The existing forms of nitrogen include nitrate (NO_3_^−^), ammonium (NH_4_^+^), and amino acids, but NO_3_^−^ is the main absorption form of crops [14]. Plant salt tolerance response is closely related to nitrogen metabolism-related enzymes. For example, nitrate reductase (NR) is the first inducible enzyme in NO_3_^–^ assimilation, and its activity is sensitive to environmental conditions [15]. GS and GOGAT activities reflect the level of nitrogen metabolism and the ability of protein synthesis [16]. Under salt stress, because NR activity may be inhibited, NO_3_^−^ reduction and NH_4_^+^ assimilation is blocked, which reduces GS and GOGAT activities and leads to NH_4_^+^ accumulation and toxic effects on plants [17]. GDH may play a unique physiological role in releasing many NH_4_^+^ under salt stress [18]. Salt stress inhibited the absorption of NO_3_^–^ and NH_4_^+^ and the transportation from root to leaf, thus affecting the activities of enzymes related to nitrogen metabolism [19].

Under salt stress, there are two kinds of substances involved in plant osmotic regulation: and one is inorganic ions, such as Na^+^, K^+^, Cl^−^, Ca^2+^, and the other is the organic substances synthesized in cells, such as betaine, proline, soluble sugar, and soluble protein [20]. Previous studies showed that with the increase of salt concentration and the extension of stress time, the free proline, soluble sugar, and soluble protein of plant seedling leaves generally increased first and then decreased [21,22,23]. Soybeans with strong salt tolerance generally have a high content of osmotic regulators [24]. The osmoregulation substances in salt-tolerant soybeans increased significantly [25]. The leaves of salt-sensitive rice varieties accumulate more soluble sugar and protein than salt-tolerant varieties under salt stress [26]. Plant hormones play an essential role in mediating the response of plants to abiotic stress. Abiotic stress often leads to changes in the production, distribution, or signal transduction of stress hormones [27]. Furthermore, the perception of stress signals triggers the signal transduction cascade in plants, and plant hormones can be used as sensors [28]. ABA can enable plants to survive under adverse environmental conditions. Due to different stresses that tend to induce ABA synthesis, it is always considered to be a plant stress hormone [29]. Abiotic stress, such as salt stress, can change the metabolism and distribution of IAA. In addition, ROS production in response to abiotic stress may also affect the metabolism of IAA [18]. Studying the salt-stress tolerance mechanism of crops, breeding salt-tolerant germplasm resources, and tapping the production potential of salinized cultivated land plays a significant role in maintaining maize grain yield increase and stable yield and ensuring food security [30]. The purpose of this study was to clarify the responses of antioxidant enzyme systems, nitrogen metabolism, osmotic regulation, and hormone regulation of inbred maize lines, so as to provide technical support and theoretical basis for the cultivation of salt-tolerant maize varieties.

## 2. Results

### 2.1. Contents of O_2_^−^, H_2_O_2_ Content, MDA, and EL Value

As shown in Figure 1, the O_2_^−^ content of B46 and NC236 increased with the increase of NaCl concentration, and the increase range of O_2_^−^ content of NC236 was greater than that of B46. When the salt concentration reached 110 mmol L^−1^, the O_2_^−^ content difference between the two maize varieties was significant. Compared with control, the O_2_^−^ content of B46 increased by 8.41%, 26.95%, 36.02%, and 45.68% under the conditions of 55, 110, 165, and 220 mmol L^−1^ NaCl. NC236 increased by 7.86%, 28.47%, 44.61%, and 53.47%. The H_2_O_2_ content of B46 and NC236 increased with the increase of NaCl concentration, and the increase range of H_2_O_2_ content of NC236 was greater than that of B46. When the salt concentration reached 110 mmol L^−1^, there was a significant difference in H_2_O_2_ content between the two inbred maize lines. Compared with control, the H_2_O_2_ content of B46 increased by 5.67%, 14.79%, 19.03%, and 28.56% under the conditions of 55, 110, 165, and 220 mmol L^−1^ NaCl. NC236 increased by 9.83%, 16.40%, 23.46%, and 29.55%. The MDA content of B46 and NC236 increased with the increase of NaCl concentration, and the increase range of MDA content of NC236 was greater than that of B46. When the salt concentration reached 110 mmol L^−1^, the MDA content difference between the two inbred maize lines was significant. Compared with control, the MDA content of B46 increased by 15.86%, 29.01%, 43.53%, and 59.67% under the conditions of 55, 110, 165, and 220 mmol L^−1^ NaCl. NC236 increased by 23.02%, 40.04%, 52.78%, and 76.51%. Under 0 mmol L^−1^ and 55 mmol L^−1^ NaCl stress, the MDA content of the two inbred maize lines did not reach a significant level. With the increase of NaCl concentration, the MDA content between the two maize seedlings reached a significant level (*p* < 0.05). The EL of B46 and NC236 increased with the increase of NaCl concentration, and the increase range of EL of NC236 was greater than that of B46. When the salt concentration reached 110 mmol L^−1^, there was a significant difference in EL content between the two inbred maize lines. Compared with control, the EL of B46 increased by 18.35%, 58.01%, 104.10%, and 167.96% under the conditions of 55, 110, 165, and 220 mmol L^−1^ NaCl. NC236 increased by 18.96%, 81.58%, 141.93%, and 181.31% (Figure 1).

### 2.2. Activities of SOD, POD, and CAT

As shown in Figure 2, the SOD activity of B46 and NC236 increased first and then decreased with the increase of NaCl concentration. B46 SOD activity began to decrease when the concentration of NaCl was 165 mmol L^−1^. When the NaCl concentration of NC236 was 110 mmol L^−1^, the SOD activity began to decrease. Compared with control, when the concentration of NaCl was 55 mmol L^−1^, the SOD activity of B46 and NC236 increased by 35.62% and 16.90%, respectively, and the increase of SOD activity of B46 was greater than that of NC236. When the concentration of NaCl was 220 mmol L^−1^, the SOD activity of B46 and NC236 decreased by 8.27% and 26.04%, respectively. The decrease of SOD activity of B46 was less than that of NC236.The POD activity of B46 and NC236 decreased with the increase of NaCl concentration. When the concentration of NaCl was 165 mmol L^−1^, the activity of POD began to decrease. When the NaCl concentration of NC236 was 55 mmol L^−1^, the POD activity began to decrease. Compared with control, the POD activity of B46 decreased by 7.33%, 12.17%, and 21.82% under the conditions of 110, 165, and 220 mmol L^−1^ NaCl. Compared with control, the POD activity of NC236 decreased by 9.27%, 16.40%, 25.47%, and 34.25% under the conditions of 55, 110, 165, and 220 mmol L^−1^ NaCl. In each NaCl concentration treatment, the decrease of POD activity of B46 was less than that of NC236, and there was a significant difference between B46 and NC236 when the NaCl concentration was 165 and 220 mmol L^−1^ (*p* < 0.05). The CAT activity of B46 and NC236 decreased with the increase of NaCl concentration, but there was no significant difference between B46 and NC236 in each NaCl concentration treatment (*p* < 0.05). Compared with control, the CAT activity of B46 decreased by 16.86%, 21.28%, 34.01%, and 41.43% under the conditions of 55, 110, 165, and 220 mmol L^−1^ NaCl. CAT activity of NC236 decreased by 16.54%, 26.77%, 38.76%, and 48.24% (Figure 2).

### 2.3. Enzyme Activities of AsA-GSH Cycle

As shown in Figure 3, the APX activity of B46 and NC236 decreased with the increase of NaCl concentration. The APX activity of B46 and NC236 began to decrease when the NaCl concentration was 110 mmol L^−1^. Compared with control, the APX activity of B46 decreased by 11.01%, 21.53%, and 37.91% under the conditions of 110, 165, and 220 mmol L^−1^ NaCl. The APX activity of NC236 decreased by 25.51%, 29.93%, and 50.09%. When NaCl concentrations were 110, 165, and 220 mmol L^−1^, there was a significant difference in APX activity between B46 and NC236 (*p* < 0.05). The GR activity of B46 and NC236 increased first and then decreased with the increase of NaCl concentration, and reached the maximum when the NaCl concentration was 55 mmol L^−1^. When NaCl concentrations were 55, 165, and 220 mmol L^−1^, there was a significant difference in GR activity between B46 and NC236 (*p* < 0.05). Compared with control, the GR activities of B46 and NC236 increased by 14.24% and 6.09%, respectively, under 55 mmol L^−1^ NaCl. Under the conditions of 165 and 220 mmol L^−1^ NaCl, the GR activity of B46 decreased by 13.35% and 40.55%, respectively, and that of NC236 decreased by 27.19% and 51.86%, respectively. The MDHAR activity of B46 and NC236 decreased with the increase of NaCl concentration. Compared with control, the MDHAR activity of B46 decreased by 6.23%, 10.76%, 18.18%, and 26.14% under the conditions of 55, 110, 165, and 220 mmol L^−1^ NaCl. The MDHAR activity of NC236 decreased by 9.12%, 14.13%, 23.25%, and 31.42%. When the concentration of NaCl was 220 mmol L^−1^, there was a significant difference in MDHAR activity between B46 and NC236 (*p* < 0.05). The DHAR activity of B46 and NC236 decreased with the increase of NaCl concentration. Compared with control, the DHAR activity of B46 decreased by 4.66%, 12.63%, 30.68%, and 40.60% under the conditions of 55, 110, 165, and 220 mmol L^−1^ NaCl. The APX activity of NC236 decreased by 8.53%, 19.56%, 44.41%, and 50.74%. When NaCl concentrations were 165 and 220 mmol L^−1^, there was a significant difference in DHAR activity between B46 and NC236 (*p* < 0.05) (Figure 3).

### 2.4. Contents of AsA, DHA, GSH, GSSG, Ratio of AsA/DHA and GSH/GSSG

As shown in Figure 4, the DHA content of B46 and NC236 had no significant difference in each NaCl concentration treatment, while AsA and AsA/DHA showed a downward trend with the increase of NaCl concentration. Compared with control, under the conditions of 55, 110, 165, and 220 mmol L^−1^ NaCl, the AsA content of B46 decreased by 5.31%, 22.28%, 30.80%, and 41.73%. The AsA content of NC236 decreased by 12.54%, 36.79%, 54.98%, and 66.31%. AsA/DHA of B46 decreased by 8.33%, 18.55%, 37.28%, and 49.25%. AsA/DHA of NC236 decreased by 16.69%, 28.59%, 48.61%, and 67.13%. The GSSG content of B46 and NC236 had no significant difference in each NaCl concentration treatment, while GSH and GSH/GSSG showed a downward trend with the increase of NaCl concentration. Compared with control, the GSH content of B46 decreased by 11.97%, 20.66%, 43.33%, and 51.01% under the conditions of 55 mmol L^−1^, 110 mmol L^−1^, 165 mmol L^−1^ and 220 mmol L^−1^ NaCl. The GSH content of NC236 decreased by 11.86%, 30.90%, 55.86%, and 66.21%. GSH/GSSG of B46 decreased by 2.16%, 11.07%, 26.07%, and 24.69%. GSH/GSSG of NC236 decreased by 7.72%, 15.71%, 53.36%, and 62.42%.

### 2.5. Contents of NO_3_^−^, NO_2_^−^, and NH_4_^+^

As shown in Figure 5, under 55 mmol L^−1^ NaCl stress, the NO_3_^−^ content of B46 increased significantly compared with control, but NC236 did not change significantly. With the increase of NaCl concentration, the NO_3_^–^ content of the two inbred maize lines decreased. However, in each concentration treatment, the decrease of NO_3_^−^ content of B46 was less than that of NC236. The NO_3_^−^ content of NC236 decreased by 15.14%, 36.42%, and 53.38%, respectively under 110, 165, and 220 mmol L^−1^ NaCl stress. With the increase of NaCl concentration, the NO_2_^–^ content of B46 and NC236 decreased, and after the NaCl concentration reached 55 mmol L^−1^, the NO_2_^–^ content of B46 and NC236 showed significant difference compared with control. Under 55 mmol L^−1^, 110 mmol L^−1^, 165 mmol L^−1^, and 220 mmol L^−1^ NaCl stress, the NO_2_^−^ content of B46 decreased by 9.90%, 15.67%, 26.63%, and 31.92% respectively, and that of NC236 decreased by 7.55%, 33.96%, 46.23%, and 61.32% respectively. With the increase of NaCl concentration, the NH_4_^+^ content of the two inbred lines showed an upward trend, and there was a significant difference between the NH_4_^+^ content of the two inbred lines after the NaCl concentration reached 110 mmol L^−1^. After the NaCl concentration reached 165 mmol L^−1^, the NH_4_^+^ content of the two inbred lines showed significant difference from control. Under 165 and 220 mmol L^−1^ NaCl stress, the NH_4_^+^ content of B46 increased by 25.83% and 33.45%, respectively, and that of NC236 increased by 50.43% and 64.35%, respectively.

### 2.6. Activities of NR and NiR

As shown in Figure 6, the NR activity of B46 and NC236 decreased with the increase of NaCl concentration, and the NR activity of the two inbred lines showed significant difference after the NaCl concentration reached 110 mmol L^−1^. Under 110, 165, and 220 mmol L^−1^ NaCl stress, the NR activity of B46 decreased by 1.45%, 20.78%, and 39.13%, respectively, and that of NC236 decreased by 31.90%, 45.62%, and 62.14%, respectively.

As shown in Figure 6, with the increase of NaCl concentration, the NiR activities of B46 and NC236 showed a downward trend, and after the NaCl concentration reached 110 mmol L^−1^, the NiR activities of the two inbred lines showed significant differences from their respective controls. Under 110, 165, and 220 mmol L^−1^ NaCl stress, the NiR activity of B46 decreased by 17.81%, 26.03%, and 45.34%, respectively, and that of NC236 decreased by 29.19%, 42.58%, and 68.90%, respectively.

### 2.7. Activities of GS, GOGAT, NADH-GDH, and NAD-GDH

As shown in Figure 7, with the increase of NaCl concentration, the GS activity of B46 and NC236 increased first and then decreased. The maximum GS activity of B46 appeared when the NaCl concentration was 110 mmol L^−1^, and the maximum GS activity of NC236 appeared when the NaCl concentration was 55 mmol L^−1^. When the concentration of NaCl reached 110 mmol L^−1^, there were significant differences in GS activity between the two inbred lines. Under 220 mmol L^−1^ NaCl stress, the GS activity of B46 decreased by 17.65% and that of NC236 decreased by 36.96%. With the increase of NaCl concentration, the GOGAT activity of B46 increased first and then decreased, and the maximum value appeared when the NaCl concentration was 110 mmol L^−1^. NaCl stress with concentrations of 55 and 110 mmol L^−1^ had no significant effect on the GOGAT activity of NC236. Under 220 mmol L^−1^ NaCl stress, the GOGAT activity of B46 decreased by 11.45% and that of NC236 decreased by 39.58%, respectively. With the increase of NaCl concentration at low concentration, the NADH-GDH activity of B46 and NC236 increased first and then decreased. The NADH-GDH activity of B46 and NC236 increased and reached the maximum at the NaCl concentration of 165 mmol L^−1^ and 110 mmol L^−1^, respectively. Under 220 mmol L^−1^ NaCl stress, the NADH-GDH activity of B46 decreased by 39.82%, respectively compared with control. The NADH-GDH activity of NC236 decreased by 58.44% compared with control. With the increase of NaCl concentration, the NAD-GDH activity of B46 and NC236 increased first and then decreased. The NAD-GDH activity of B46 and NC236 increased and reached the maximum at the NaCl concentration of 165 mmol L^−1^ and 110 mmol L^−1^, respectively. Under 220 mmol L^−1^ NaCl stress, the NAD-GDH activity of B46 decreased by 37.81% and that of NC236 decreased by 75.99%, respectively.

### 2.8. Activities of GOT and GPT

As shown in Figure 8, with the increase of NaCl concentration, the GOT activities of B46 and NC236 increased first and then decreased. When the NaCl concentration was 55 mmol L^−1^, the GOT activities of B46 and NC236 reached the maximum value, increasing by 12.44% and 23.18%, respectively, compared with control. There was no significant difference in GOT activity of B46 under NaCl concentration of 55 and 110 mmol L^−1^, which decreased after reaching 165 mmol L^−1^. The GOT activity of NC236 decreased since the concentration of NaCl was 110 mmol L^−1^. When the concentration of NaCl was 165 and 220 mmol L^−1^, the GOT activity of B46 decreased by 12.29% and 31.23%, respectively. The GOT activity of NC236 decreased by 19.26% and 50.42%, respectively. When the concentration of NaCl ranged from 0 to 110 mmol L^−1^, the GPT activity of B46 did not change significantly. The concentration of NaCl continued to increase, and the GPT activity of B46 decreased. The GPT activity of NC236 increased first and then decreased with the increase of NaCl concentration, and the maximum value appeared in the treatment with NaCl concentration of 55 mmol L^−1^. There were significant differences between the two inbred maize lines at 55, 165, and 220 mmol L^−1^. When the concentration of NaCl reached 165 mmol L^−1^, the GPT activities of B46 and NC236 were significantly different from those of control. When the concentration of NaCl was 165 and 220 mmol L^−1^, the GPT activity of B46 decreased by 17.27% and 34.01%, respectively. The GPT activity of NC236 decreased by 36.82% and 42.82%, respectively (Figure 8).

### 2.9. Contents of Proline, Soluble Protein, and Soluble Sugar

As shown in Figure 9, the proline content in seedling leaves gradually increased with the increase of NaCl concentration. Under 55 mmol L^−1^ NaCl stress, the proline contents of B46 and NC236 increased by 6.48% and 0.85%, respectively, compared with control. Under 110 mmol L^−1^ NaCl stress, the proline contents of B46 and NC236 increased by 50.60% and 32.96%, respectively, compared with control, but the difference between them did not reach a significant level. Under 165 and 220 mmol L^−1^ NaCl stress, the proline content of B46 and NC236 increased significantly. The increase range of B46 was greater than that of NC236, increased by 110.87% and 174.52%, respectively. 55.63% and 98.38%, which reached a significant level compared with control (*p* < 0.05). The soluble protein content of seedling leaves increased gradually with the increase of NaCl concentration. Under 55 mmol L^−1^ NaCl stress, the soluble protein contents of B46 and NC236 increased by 6.48% and 5.85%, respectively, compared with control, but the difference was not significant. Under 110, 165, and 220 mmol L^−1^ NaCl stress, the soluble protein content of B46 and NC236 increased significantly. The increase range of B46 was greater than that of NC236, increased by 50.60%, 110.87% and 174.52%, respectively, and reached a significant level compared with control (*p* < 0.05). After salt stress treatment, the soluble sugar content of leaves of two maize seedlings increased, and the greater the NaCl stress intensity, the more significant the increase of soluble sugar content. Under 55, 110, 165, and 220 mmol L^−1^ NaCl stress, the soluble sugar content of B46 increased by 7.11%, 28.39%, 58.89%, and 87.59%, respectively, compared with control, and the soluble sugar content of NC236 increased by 5.58%, 18.81%, 42.66%, and 69.72%, respectively. The increase of B46 was greater than that of NC236. Under 55 mmol L^−1^ NaCl stress, it did not reach a significant level compared with control. When the NaCl concentration reached 110 mmol L^−1^, the soluble sugar content among inbred maize lines reached a significant level (*p* < 0.05) (Figure 9).

### 2.10. Contents of Na^+^, K^+^, and K^+^/Na^+^ Ratio

It can be seen from Table 1 that with the increase of NaCl concentration, the Na^+^ content in the leaves of the two inbred maize lines increased gradually, and the increase range of Na^+^ content of NC236 was greater than that of B46. Under 55, 110, 165, and 220 mmol L^−1^ NaCl stress, the Na^+^ content of B46 increased by 1.06, 1.89, 2.59, and 3.59 times, respectively compared with control. When the NaCl concentration is greater than 55 mmol L^−1^, the Na^+^ content of NC236 reaches a significant level compared with control. When the NaCl concentration is greater than 110 mmol L^−1^, the Na^+^ content of B46 reaches a significant level compared with control, and there is also a difference between the two inbred maize lines.

The K^+^ content of leaves of two maize seedlings decreased gradually after salt stress, and the decrease range of B46 was less than that of NC236. Under 55, 110, 165, and 220 mmol L^−1^ NaCl stress, the K^+^ content of B46 decreased by 3.59%, 6.93%, 9.79%, and 12.59%, respectively, and the K^+^ content of NC236 decreased by 4.74%, 7.78%, 13.49%, and 18.10%, respectively. When the salt concentration reached 220 mmol L^−1^, there was a difference between the two inbred maize lines. With the gradual increase of NaCl concentration, the K^+^/Na^+^ ratio of the two inbred maize lines showed a gradual downward trend, and the downward trend of NC236 was higher than that of B46. Under 55, 110, 165, and 220 mmol L^−1^ NaCl stress, the K^+^/Na^+^ ratio of B46 decreased by 53.64%, 67.98%, 75.05%, and 79.75% respectively compared with control, and the K^+^/Na^+^ ratio of NC236 decreased by 54.74%, 69.55%, 78.68%, and 82.44% compared with control (Table 1).

### 2.11. Contents of IAA, Z + ZR, and ABA

Appendix A showed that 55 and 110 mmol L^−1^ NaCl had no significant effect on the content of IAA in the roots of maize seedlings. Under salt stress, the increase of IAA content in NC236 was greater than that in B46. Under 165 and 220 mmol L^−1^ NaCl stress, the content of IAA in NC236 roots of B46 increased by 22.43% and 51.23%, respectively, compared with control. The content of IAA in NC236 roots increased by 44.41% and 72.95%, respectively, compared with control. It can be seen from Appendix A that there is no significant difference between the content of IAA in B46 and NC236 leaves compared with that in control, but the ratio of IAA in NC236 leaves is always significantly lower than that in B46 under salt stress. Salt stress significantly reduced the content of IAA in B46 and NC236 leaves and had a more obvious effect on the content of IAA in NC236 leaves. Under 55, 110, 165, and 220 mmol L^−1^ NaCl treatment, the content of IAA in B46 leaves decreased by 15.26%, 25.43%, 12.25%, and 46.16%, respectively. The content of IAA in NC236 leaves decreased by 45.37%, 50.49%, 39.20%, and 71.74%, respectively. The content of Z + ZR in NC236 roots is always lower than that in B46. NaCl treatment significantly reduced the content of Z + ZR and had a more significant effect on NC236. Compared with control, the content of Z + ZR in B46 roots decreased by 15.06%, 22.21%, 17.19%, and 38.22% respectively under 55, 110, 165, and 220 mmol L^−1^ NaCl treatment. Compared with control, the content of Z + ZR in NC236 roots decreased by 19.13%, 34.98%, 25.49%, and 48.91% respectively. As can be seen from Appendix A, the content of Z + ZR in NC236 leaves is always lower than that in B46. NaCl treatment significantly reduced the content of Z + ZR and had a more significant effect on NC236. Compared with control, the content of Z + ZR in B46 leaves under 55, 110, 165 and 220 mmol L^−1^ NaCl treatment decreased by 17.34%, 29.30%, 24.76%, and 43.85% respectively. The content of Z + ZR in NC236 leaves decreased by 25.90%, 45.84%, 37.95%, and 54.16%, respectively. The content of ABA in NC236 roots is always higher than that in B46 roots. With the increase of NaCl concentration, the content of ABA in the roots of the two inbred maize lines increased gradually, and the increase of ABA content in NC236 roots was greater than that in B46 roots. Compared with control, the content of ABA in B46 roots increased by 24.95%, 55.59%, 52.15%, and 128.95%, respectively, under 55, 110, 165, and 220 mmol L^−1^ NaCl treatment. The content of ABA in NC236 roots increased by 74.07.11%, 114.67.01%, 91.32%, and 196.05%, respectively. As can be seen from Appendix A, under the control condition, there is no significant difference between the content of ABA in B46 and NC236 leaves, while under the NaCl treatment condition, the ratio of ABA in B46 leaves is always significantly lower than that in NC236 leaves. Salt stress significantly reduced the content of ABA in B46 and NC236 leaves, and the change of ABA content in NC236 leaves was more obvious. Compared with control, under 55, 110, 165, and 220 mmol L^−1^ NaCl treatment, the content of ABA in B46 leaves increased by 24.95%, 55.59%, 52.15%, and 90.96%, respectively. The content of ABA in NC236 leaves increased by 73.71%, 114.22%, 90.92%, and 146.42% respectively compared with control (Appendix A).

### 2.12. Ratio of IAA/(Z + ZR) and IAA/ABA

It can be seen from Appendix A that the ratio of IAA/(Z + ZR) in NC236 roots is always higher than that in B46 roots. An amount of 55 mmol L^−1^ NaCl had no significant effect on the ratio of IAA/(Z + ZR) in B46 and NC236 roots. With the increase of NaCl concentration, the ratio of IAA/(Z + ZR) in the roots of the two inbred maize lines gradually increased, and the increase of IAA/(Z + ZR) ratio in NC236 roots was greater than that in B46 roots. Compared with control, the ratio of IAA/(Z + ZR) in B46 roots increased by 29.92%, 46.63%, and 143.24%, respectively, under 110, 165, and 220 mmol L^−1^ NaCl treatment. The ratio of IAA/(Z + ZR) in NC236 roots increased by 65.28%, 94.96%, and 237.54%, respectively. Appendix A showed that there is no significant difference between the ratio of IAA/(Z + ZR) in B46 and NC236 leaves under control conditions, while the ratio of IAA/(Z + ZR) in NC236 leaves is always significantly lower than that in B46 under salt stress. Salt stress significantly decreased the ratio of IAA/(Z + ZR) in B46 and NC236 leaves and had a more obvious effect on the ratio of IAA/(Z + ZR) in NC236 leaves. Under 55, 110, 165, and 220 mmol L^−1^ NaCl treatment, the ratio of IAA/(Z + ZR) in B46 leaves decreased by 17.26%, 16.52%, 7.72%, and 23.81%, respectively. The ratio of IAA/(Z + ZR) in NC236 leaves decreased by 37.75%, 40.52%, 31.49%, and 53.74%, respectively. There is no significant difference in the ratio of IAA/ABA between B46 and NC236 roots under control conditions, while the ratio of IAA/ABA in NC236 roots is always significantly lower than that in B46 under salt stress. Salt stress significantly reduced the IAA/ABA ratio in B46 and NC236 roots and had a more obvious effect on the IAA/ABA ratio in NC236. Under 55, 110, 165, and 220 mmol L^−1^ NaCl treatment, the ratio of IAA/ABA in B46 roots decreased by 40.69%, 58.29%, 49.51%, and 75.51%, respectively, compared with control. The ratio of IAA/ABA in NC236 roots decreased by 68.13%, 76.68%, 67.54%, and 88.40%, respectively. Appendix A showed that under control condition, there is no significant difference in the ratio of IAA/ABA in B46 and NC236 leaves, while under the NaCl treatment condition, the ratio of IAA/ABA in NC236 leaves is always significantly lower than that in B46 leaves. Salt stress significantly reduced the IAA/ABA ratio in B46 and NC236 leaves and had a more obvious effect on the IAA/ABA ratio in NC236 leaves. Under 55, 110, 165, and 220 mmol L^−1^ NaCl treatment, the ratio of IAA/ABA in B46 leaves decreased by 20.41%, 36.54%, 21.21%, and 34.32%, respectively. The ratio of IAA/ABA in NC236 leaves decreased by 46.29%, 55.97%, 40.21%, and 53.75%, respectively.

## 3. Discussion

Growth inhibition is the most sensitive expression of plants to salt stress [31]. Plants always respond to the adverse environment by changing their growth and morphological characteristics, such as reducing plant height, root length, fresh weight, and dry weight [32]. The production and clearance of O_2_^−^ in plant cells are in a dynamic balance. Salt stress can result in a considerable accumulation of O_2_^−^ and further damage to plant cell structure [33]. Salt stress can significantly inhibit the SOD activity of wheat and alfalfa, and the SOD activity increases first and then decreases [10,34]. In this study, SOD activity increased under low concentration NaCl (55 mmol L^−1^) stress. The reason may be that low NaCl stress activates the antioxidant system in plants. Nevertheless, the O_2_^−^ content in seedling leaves increased, which may be because the O_2_^−^ production rate is greater than the sod clearance rate under stress. With the increase of NaCl concentration, SOD activity showed a downward trend, which may be due to the destruction of an antioxidant system caused by the high concentration of salt, resulting in the decline of enzyme activity and the decline of defense mechanism against salt damage. The increase of O_2_^−^ content in B46 was less than that in NC236, which may be caused by the difference of SOD activity between two inbred maize lines with different salt tolerance. In this study, with the increase of NaCl concentration, the H_2_O_2_ content of B46 and NC236 leaves also increased, which may be due to the large production of O_2_^−^ and the increase of substrate of SOD catalytic reaction under salt stress, resulting in the increase of H_2_O_2_ content. However, the increase of H_2_O_2_ content in B46 is less than that in NC236, due to at least two following reasons. Firstly, compared with NC236, B46 seedling leaves produce less H_2_O_2_. Secondly, compared with NC236, B46 seedling leaves had a stronger H_2_O_2_ scavenging ability. Previous studies have shown that with the gradual increase of NaCl concentration, the O_2_^−^ production rate and H_2_O_2_ content in leaves show a significant increase trend [35,36]. In this study, the increase of O_2_^−^ and H_2_O_2_ content can explain the inhibition of seedling growth under salt stress, while the increase of O_2_^−^ and H_2_O_2_ content of B46 is less than that of NC236 under the same salt stress. The difference in O_2_^−^ and H_2_O_2_ contents between B46 and NC236 can partly explain the difference in biomass under salt stress.

The unsaturated fatty acid is one of the crucial components of the plant cell membrane, which is vulnerable to ROS attack, resulting in biofilm damage and a significant accumulation of membrane lipid peroxide [37]. MDA content and EL are critical indicators to measure the degree of plasma membrane injury under stress [38]. With the gradual increase of salt stress concentration, the permeability of plant plasma membrane will gradually increase, destroy the structure of biofilm, and finally lead to the increase of MDA content [39]. The MDA content of cucumber seedling leaves under salt stress was significantly higher than control [10]. In this study, with the increase of NaCl concentration, MDA content and EL in maize seedling leaves increased. The increase of MDA content and EL in B46 was less than NC236 under the same salt stress, which showed that the content of O_2_^−^ and H_2_O_2_ in maize seedling leaves increased, causing oxidative stress and damage to cell membrane, and the solute in the cell penetrated from the cell into the external environment. However, the damage of NC236 to cell membrane was more incredible than B46. Previous studies have shown that the activities of POD and CAT in maize leaves gradually increase with the increase of NaCl concentration, while the peak of POD activity in salt-tolerant lines appears later [40]. In this study, the POD and CAT activities of B46 and NC236 showed a downward trend with the increase of NaCl concentration. However, the POD activity of B46 was significantly higher than that of NC236, but there was no significant difference in CAT activity between B46 and NC236, which may be because CAT enzyme was not the main factor leading to the difference in antioxidant stress ability between B46 and NC236. AsA and GSH, together with APX, DHAR, MHDAR, GR, and other antioxidant enzymes, constitute a circulating system that can effectively scavenge free radicals, namely the AsA-GSH cycle. The former study showed that the contents of AsA and GSH in rice with weak salt tolerance decreased significantly under salt stress [41]. High salt concentration treatment significantly reduced the content of AsA and increased the content of GSH in mung bean [42]. In this study, with the increase of NaCl concentration, the contents of AsA and GSH, AsA/DHA, and GSH/GSSG in maize seedlings decreased, but the contents of DHA and GSSG did not change significantly. Compared with NC236, the contents of AsA and GSH, AsA/DHA and GSH/GSSG of B46 decreased slightly. The leaf cells of B46 maintained a high content of AsA, so that APX had enough substrate to promote the clearance of H_2_O_2_. In addition, higher GSH concentration can effectively reduce—S-S bond, stabilize—SH group and stabilize the structure of membrane proteins. Previous studies have shown that MDHAR and DHAR activities decrease, and APX and GR activities increase under salt stress [43]. In this study, with the increase of NaCl concentration, the activities of APX, DHAR, and MHDAR of B46 and NC236 decreased continuously, and the activity of GR increased first and then decreased. Compared with NC236, the activities of APX, DHAR, and MHDAR of B46 decreased slightly. In 55 mmol L^−1^ NaCl treatment, the increase of GR activity of B46 was greater than that of NC236. The results showed that compared with NC236, the four enzymes APX, DHAR, MHDAR, and GR of B46 coordinated with each other, so that B46 maintained a relatively high H_2_O_2_ scavenging capacity, enhanced the antioxidant capacity to reduce the damage of reactive oxygen species to plant cell membrane, inhibited the increase of membrane lipid peroxidation and conductivity, and reduced the damage degree of plant cells by reactive oxygen species.

NO_3_^−^ enters the root and is transported to the blade for a long distance. NO_3_^−^ is the only storage form of nitrogen. To synthesize amino acids, it is catalyzed by nitrate reductase (NR) and nitrite reductase (NiR). Under high concentration salt stress, the NR and NiR activities in leaves of salt-sensitive buckwheat varieties decreased significantly, while the decreased range of salt-tolerant buckwheat varieties was relatively small [44]. However, studies also showed that under NaCl stress, NO_3_^−^ content in roots and leaves of ophiopogon japonicus showed an upward trend [45]. In this study, under the stress of low concentration NaCl (55 mmol L^−1^), the NO_3_^−^ content increased, which may be because low concentration NaCl promoted root growth and NO_3_^−^ absorption. With the increase of NaCl concentration, the NO_3_^−^ content in maize leaves increased first and then decreased. This may be due to the fact that NaCl stress inhibits the absorption of NO_3_^−^ by roots or the long-distance transportation of NO_3_^−^ from roots to leaves. The decrease of NO_3_^−^ content in leaves of NC236 is greater than that of B46, which may be because the roots of salt-tolerant inbred maize lines maintain higher root biomass under NaCl stress, which is conducive to maintaining NO_3_^−^ absorption. At the same time, salt-tolerant inbred maize lines have greater transportation rate, which is conducive to maintaining the long-distance transportation of NO_3_^−^ from root to leaf, and finally shows that the decrease of NO_3_^−^ content is negligible. Under salt stress, NR activity decreased with the accumulation of substrate NO_3_^−^ and the decrease of product NO_2_^−^ [46]. NR activity was mainly affected by NO_3_^−^ concentration. In this study, under NaCl stress, the content of NO_2_^−^ also decreased due to the decrease of NO_3_^−^ content and NR activity. Under the same stress conditions, the decrease of NR activity in NC236 leaves was more significant than that in B46, which may be related to the difference of NO_3_^−^ content between the two leaves. Consistent with the change law of NR activity, NiR activity decreased significantly under drought stress, which may be related to the decrease of NO_3_^−^ and NO_2_^−^ content under salt stress. Salt-tolerant inbred maize lines maintained high NiR activity under NaCl stress, and then maintained the transformation of NO_2_^−^ to NH_4_^+^ to a certain extent.

The GS/GOGAT cycle is the main NH_4_^+^ assimilation pathway for plants. In this cycle, NH_4_^+^ is converted to glutamine by glutamine synthase (GS) and then converted to glutamate by GOGAT, which is directly integrated into the structure of amino acids. With the increase of NaCl concentration, the contents of NO_3_^−^ and NO_2_^−^ in Chinese cabbage decreased significantly, but the content of NH_4_^+^ accumulated [47]. The activities of GS/GOGAT and GDH in potato continued to decline [48]. Under salt stress, the GS activity of barley increased, and there was no significant change in GDH activity [49]. In this study, the activity of GS decreased under NaCl stress. It may be related to the increased NH_4_^+^ content in leaves under salt stress. Compared with NC236, the decrease of NR and NiR activities of B46 was slight, but the increase of NH_4_^+^ content was also small. One possible explanation is that salt-tolerant inbred maize lines maintain high GS and GOGAT activities and promote the assimilation of NH_4_^+^. Under stress conditions, when GS/GOGAT pathway is inhibited, the content of NH_4_^+^ in plant cells will increase significantly. NH_4_^+^ can be used as a matrix to catalyze the formation of glutamate through GDH. In this study, there was no significant change in NADH-GDH activity and NAD-GDH activity of B46 and NC236 at a low concentration of NaCl (55 mmol L^−1^). When the NaCl concentration reached 110 mmol L^−1^, the NADH-GDH activity of NC236 and the NAD-GDH activity of the two inbred lines increased significantly compared with control, and the increase showed that NC236 was greater than B46, which may be due to the large accumulation of NH_4_^+^ in NC236 and the increase of GDH enzyme activity. When the NaCl concentration was 165 mmol L^−1^, the NADH-GDH activity and NAD-GDH activity of the two inbred lines were significantly lower than control, and the decline showed that B46 was less than NC236, indicating that the NADH-GDH activity and NAD-GDH activity of B46 had more robust tolerance to NaCl stress, and could maintain a certain degree of NH_4_^+^ assimilation ability under higher NaCl stress and reduce the toxic effect of excess ammonia.

Transamination is a vital process of nitrogen metabolism in plants, which is closely related to nitrogen flow from amide and amino groups. The results showed that the GOT and GPT activities of B46 and NC236 decreased with the increase of NaCl concentration compared with control. Since glutamate is the first amino acid synthesized by inorganic nitrogen in plants and an important reaction substrate of GOT and GPT, it is mainly formed by NH_4_^+^ catalyzed by GS/GOGAT. The decline of GOT and GPT activity in this study may be related to the weakening of the GS/GOGAT pathway under salt stress. Compared with NC236, the decrease of GOT and GPT activities of B46 was small, which may be due to the small inhibitory effect of salt stress on the GS/GOGAT pathway of B46, which maintained the progress of a series of amino transfer reactions catalyzed by GOT and GPT with glutamate as substrate.

Previous studies have shown that the salt resistance of plants under salt stress is related to the content of proline, soluble protein, and soluble sugar [50,51]. Many salt ions penetrate into plant cells, and some ions will be zoned in vacuoles, to effectively alleviate the osmotic stress caused by salt stress [52]. The soluble sugar content can reduce osmotic potential and increase water retention capacity [53]. The results showed that under the stress of low concentration NaCl (55 mmol L^−1^), the proline content, soluble sugar content, and soluble protein content of B46 and NC236 did not change significantly compared with control.Compared with NC236, B46 can enhance the osmotic regulation ability of seedlings by increasing the content of soluble protein, which is conducive to maintaining the normal morphological structure and metabolic process of seedlings under osmotic stress. With the increase of NaCl concentration, the proline content, soluble protein content, and soluble sugar content of maize seedling leaves showed an upward trend, indicating that the accumulation of osmotic regulation in plants is an adaptation to stress, which is conducive to improving plant salt tolerance [54,55]. The proline content of B46 and NC236 showed significant differences when the NaCl concentration reached 165 mmol L^−1^, and the soluble protein and soluble sugar content showed significant differences when the NaCl concentration reached 110 mmol L^−1^. The increase range of proline, soluble protein, and soluble sugar content of B46 was more significant than that of NC236, indicating that under low salt concentration stress (55 mmol L^−1^ NaCl and 110 mmol L^−1^ NaCl), the strong osmotic regulation ability of B46 is mainly attributed to the accumulation of soluble protein and soluble sugar. When the concentration of NaCl is greater than 110 mmol L^−1^ NaCl, the strong osmotic regulation ability of B46 is attributed to the accumulation of proline, soluble protein, and soluble sugar, which can alleviate the damage of environmental stress to plants.

Most salts are stored in soil in the form of absorbable ions, and Na^+^ is the main salt ion leading to soil salinization. K^+^ can be used as a catalyst for more than 60 enzymes in plant cells, and can promote the synthesis and transportation of proteins and sugars [56]. It is also an important osmotic regulatory component and a large number of elements required in plant growth and development [57,58]. In this study, with the increase of salt stress intensity, the Na^+^ content in the leaves of B46 and NC236 increased, and the K^+^ content decreased, which may be due to a large amount of Na^+^ infiltrating into the seedling roots and transporting to the seedling leaves with the transport tissue, resulting in a sharp increase in the Na^+^ content in the cells. At the same time, a large amount of K^+^ flowed out, resulting in the decrease of the K^+^/Na^+^ ratio. The smaller the ratio of K^+^/Na^+^, the greater the inhibitory effect of Na^+^ on K^+^ absorption, which destroys the original ion balance and makes plants suffer ion toxicity. Promote the expression of K^+^ channel gene and high affinity K^+^ transport system, significantly increase the intracellular K^+^ content, and discharge a large amount of Na^+^ outside the cells, significantly improving the salt tolerance of crops. Previous studies have shown that most plants have significant sodium rejection [59]. In this experiment, the leaves of maize seedlings had apparent sodium rejection under low concentration NaCl stress, but the sodium rejection was lost under a certain NaCl stress. The increase of Na^+^ content in B46 leaves was less than that of NC236, and the decrease of K^+^ content under salt stress was less than that of NC236, so B46 maintained a higher K^+^/Na^+^ value, which may be because B46 absorbed less Na^+^ and more K^+^ to maintain plant growth compared with NC236, which may be closely related to the difference of salt resistance between the two.

Changes in hormone levels mediate changes in leaf morphology, and stress environment will induce changes in plant hormone balance [60]. Under salt stress, the auxin homeostasis and distribution pattern of Arabidopsis thaliana change, which will inhibit its lateral root germination and organogenesis [61]. NaCl treatment will lead to the accumulation of ABA and the decrease of cytokinin and IAA in wheat seedlings, which is characterized by slow growth [62]. In this study, salt stress caused significant changes in endogenous plant hormones in maize seedlings, and there were differences in hormone changes in roots and leaves. The contents of IAA and Z + ZR in the leaves of maize seedlings decreased, the content of ABA increased, and the change of plant hormone level in the leaves finally led to the slowdown of leaf growth and concentrated energy on the metabolism related to salt resistance [63]. IAA and Z + ZR can promote root growth, while ABA synthesis in root tips and aging leaves can inhibit plant growth [64]. The content of IAA and ABA in roots increased and the content of Z + ZR decreased, indicating that IAA in the three hormones plays a leading role in maize root growth under salt stress. Root growth and the absorption of water and nutrients are promoted, and the relatively stable metabolic process of plants under stress is maintained. The difference of endogenous hormone levels between the two inbred lines made them have different responses to salt stress in root and leaf morphology, resulting in the difference of salt tolerance.

## 4. Materials and Methods

### 4.1. Plant Materials and Growth Condition

Based on the results of preliminary experiments, we selected 242 maize inbred lines in the preliminary experiment, including 9 TS germplasm resources, 99 SS germplasm resources, 1 POP germplasm resources, 103 NSS germplasm resources, and 30 MIXED germplasm resources. We conducted hydroponic experiments and pot experiments, based on salt tolerance coefficient and correlation coefficient analysis, principal component analysis, and comprehensive evaluation of salt tolerance. Based on the screening results, we found that B46 and NC236 had great differences in salt resistance. Therefore, B46 (salt-tolerant inbred line) and NC236 (salt-sensitive inbred line) were selected as the experimental material to better explain the difference characteristics of salt resistance. After disinfection treatment, the maize seeds were put into a petri dish covered with double-layer filter paper and germinated in the dark in a 24–26 °C incubator. After 5 d, the seedlings with uniform growth were planted on a foam board with a uniform round hole and then placed in a container filled with 1/2 Hoagland nutrient solution. The seedlings were treated with salt stress when they grew to 3 leaves. The nutrient solution was changed every 2 days, and continuous ventilation was performed. The light intensity, relative humidity, light hour, and temperature were 400 μmol m^−2^ s^−1^, 70%, 13/11 h and 25/18 °C (day/night), respectively. The NaCl concentration gradients were 55 mmol L^−1^, 110 mmol L^−1^, 165 mmol L^−1^ and 220 mmol L^−1^, and the water treatment was used as the control. Samples were taken as experimental samples after 7 days of NaCl stress.

### 4.2. Determination of O_2_^−^ Content and H_2_O_2_ Content

O_2_^−^ content is determined according to the following method. A total of 0.5 g fresh seedling leaves was added to 1.0 mL of 0.05 mol L^−1^ phosphoric acid buffer (pH 7.8) and a little quartz sand into the mortar, 10,000 g for 10 min at 4 °C. Then, 0.5 mL of supernatant was added, 0.5 mL of phosphoric acid buffer, and 1 mL of 10 mmol L^−1^ hydroxylamine hydrochloride, standing at 25 °C for 1 h. Then, 1 mL of 17 mmol L^−1^ p-Aminobenzene sulfonic acid and 1 mL of 7 mmol L^−1^ α- Naphthylamine were added, standing at 25 °C for 20 min, colorimetric at 530 nm [65].

The H_2_O_2_ content was determined according to the following method. A total of 0.5 g fresh seedling leaves was added 5 mL of 0.1% TCA, ground to homogenate, and 12,000 at 4 °C × G centrifugation for 20 min; 1 mL supernatant was taken for analysis and determination [66].

### 4.3. Determination of MDA Content and Relative Conductivity (EL)

MDA content was determined by thiobarbituric acid (TBA) colorimetry. Take 0.5 g of fresh seedling leaves, add a small amount of phosphoric acid buffer (pH 7.8), grind them into homogenate, fix the volume to 5 mL, and put them into 10,000 at 4 °C × G centrifuged for 15 min. Take 1 mL of enzyme solution, add 2 mL of 10% trichloroacetic acid containing 0.6% thiobarbituric acid, place it in boiling water bath for 10 min, take the supernatant, and determine the absorbance values of 450 nm, 532 nm, and 600 nm [67].

The relative conductivity (EL) is determined according to the following method. Select fresh seedling leaves and take leaf discs with a punch (1 cm in diameter), take 10 pieces for each treatment, wash them with distilled water, put them into a 10 mL centrifuge tube for constant volume, keep them away from light for 24 h, measure the leaf conductivity with DDS-11A conductivity meter (EC1), take seedling samples in boiling water bath for 20 min, cool them to room temperature, and measure the conductivity (EC2). The relative conductivity formula is EL (%) = (EC1/EC2) × 100 [32].

### 4.4. Determination of SOD, POD, CAT, APX, GR, MDHAR and DHAR Activities

The activities of SOD, POD, and CAT were determined by nitrogen blue tetrazole method, guaiacol method, and potassium permanganate titration [68]. The activities of APX, GR, MDHAR, and DHAR were determined by referring to previous literature [69,70].

### 4.5. Determination of AsA, DHA, GSSG, GSH, NO_3_^−^, NO_2_^−^ and NH_4_^+^ Contents

The contents of AsA, DHA, GSSG, and GSH were determined with reference to the methods of Hodges and Griffith [68,71]. NO_3_^−^ content is determined according to Cataldo’s method [72]. NO_2_^−^ content is determined according to Barro et al. [73]. NH_4_^+^ content was determined according to the method of BR ä utigam [56].

### 4.6. Determination of NR, NiR, GS, GOGAT, GDH, GOT and GPT Activities

For the determination of NR activity, refer to the method of Barro et al. (1991) [73]. NiR activity was determined according to the method of Ida and Morita (1973) [74]. For the determination of GS activity, refer to the method of O’Neal and Joy (1973) [75]. The activities of GOGAT and GDH were determined according to the methods of Groat and Vance [76]. GOT and GPT activities were determined according to Liang’s method [77].

### 4.7. Determination of Osmoregulation Substance Contents

Determination of soluble protein content: weigh 0.5 g of fresh seedling leaves, add 5 mL of 3% sulfosalicylic acid into 10 mL centrifuge tube, boil water bath in water bath for 10 min, cool and filter, absorb 2 mL of filtrate, 3 mL of acid ninhydrin and 2 mL of glacial acetic acid, shake well, boil water bath for 40 min, cool to room temperature, add 5 mL of toluene to the solution, shake fully, and static stratification, absorb the upper solution into the cuvette, and measure the absorbance value at 520 nm [78].

Soluble protein content: determined by Coomassie brilliant blue G-250 method. Accurately weigh 0.5 g of fresh seedling leaves, add 2 mL distilled water and a small amount of quartz sand in the mortar to grind them into homogenate, fix the volume of distilled water to 5 mL, centrifuge at 10,000 R/min, and take the supernatant for standby. Put 0.1 mL supernatant into the test tube, add 5 mL G-250 reagent, stand for 5 min, compare the color at 520 nm [79].

Soluble sugar content: accurately weigh 0.5 g of fresh seedling leaves, add 25 mL distilled water into 50 mL triangular flask, boil water bath for 30 min, cool to room temperature, rinse the residue with hot water for several times, and fix the volume in 100 mL volumetric flask. Add 0.5 mL of extract, 0.5 mL of distilled water and 5 mL of anthrone reagent to the test tube, shake well, boiling water bath for 15 min, cool to room temperature, compare the color at 620 nm [80].

### 4.8. Determination of Na^+^ and K^+^ Contents

Dry the aboveground part of the seedling, accurately weigh 0.5 g of the sample after crushing, after digestion with concentrated sulfuric acid hydrogen peroxide, and determine the content of Na^+^ and K^+^ with M410 flame photometer [81].

### 4.9. Determination of Hormone Contents

Auxin (IAA), zeatin (Z), zeatin nucleoside (ZR), and abscisic acid (ABA) were determined by ELISA. Take 0.1 g of fresh leaves (the third piece from the top) and 0.1 g of roots of each treated plant as samples. The samples were ground in an ice bath in 80% (*v*/*v*) methanol containing 1 mm butyl hydroxyanisole (BHT) as an antioxidant. After extraction at 4 °C for 4 h, the extract was centrifuged at 12,000 g and 4 °C for 15 min. The resulting supernatants were mixed together and passed through a C18 column (C18 SEP Park cartridge, waters crop, Millford, MA, USA). After filtration, the hormone components were dried with nitrogen and dissolved in phosphate buffered saline (PBS, pH 7.5) containing 0.1% (*v*/*v*) Tween 20 and 0.1% (*w*/*v*) gelatin, and then analyzed by ELISA. IAA, Z, ZR, and ABA hormone determination kits were provided by the crop chemical regulation research center of China Agricultural University [82].

### 4.10. Data Analysis

The data were expressed by the measured mean value, analyzed by SPSS19.0 (IBM SPSS Statistics, 2010), and compared by Duncan’s new complex difference method (α = 0.05), and origin 8 is used for drawing.

## 5. Conclusions

Compared with NC236, B46 had a smaller decrease in antioxidant enzyme activity, AsA and GSH contents, AsA/DHA, and GSH/GSSG, maintained lower O_2_^−^, H_2_O_2_ contents in maize seedling leaves, further inhibited the increase of MDA content and EL, reduced the degree of peroxidation, and maintained regular metabolism. Under salt stress, compared with NC236, B46 can maintain the normal progress of NO_3_^−^ assimilation, NH_4_^+^ assimilation, and transamination to a certain extent. Under salt stress, compared with NC236, B46 synthesized osmoregulatory substances (proline content, soluble protein content, and soluble sugar) more rapidly to maintain cell turgor and make the metabolism in the body normal and orderly. Compared with NC236, B46 has a strong sodium rejection effect, maintains the absorption of K^+^, and inhibits the impact of ion toxicity on the body. Under NaCl stress, the contents of IAA in leaves and Z + ZR in leaves and roots of maize seedlings decreased, while the contents of IAA in roots and ABA in roots and leaves increased. Under salt stress, compared with NC236, B46 had higher IAA content in leaf, higher Z + ZR content in leaf and root, and lower ABA content in leaf and root. Under salt stress, compared with NC236, B46 can effectively change the adaptive morphological structure of seedling roots and leaves through the change of hormone level, so as to reduce the negative effect of salt stress on seedling growth.

## Figures and Tables

**Figure 1 plants-11-01547-f001:**
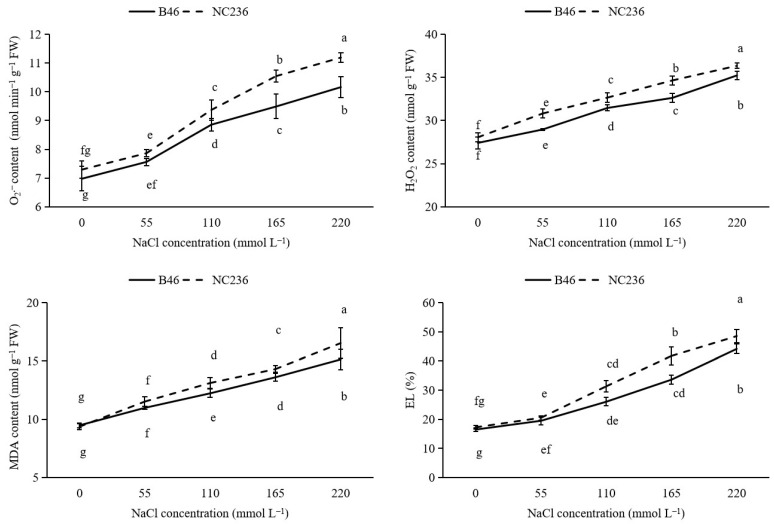
Effect of salt stress on the contents of O_2_^−^, H_2_O_2_, MDA, and EL value in maize leaves. Data are expressed as mean ± standard deviation. Different letters within the same column indicate significant difference at the 5% level.

**Figure 2 plants-11-01547-f002:**
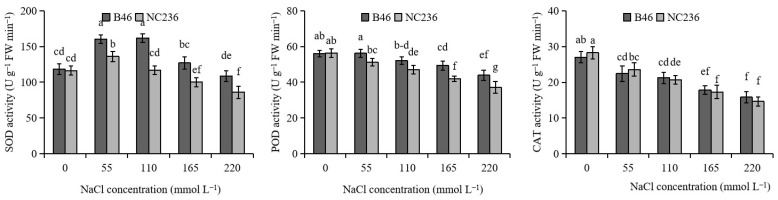
Effect of salt stress on the activities of SOD, POD, and CAT in maize leaves. Data are expressed as mean ± standard deviation. Different letters within the same column indicate significant difference at the 5% level.

**Figure 3 plants-11-01547-f003:**
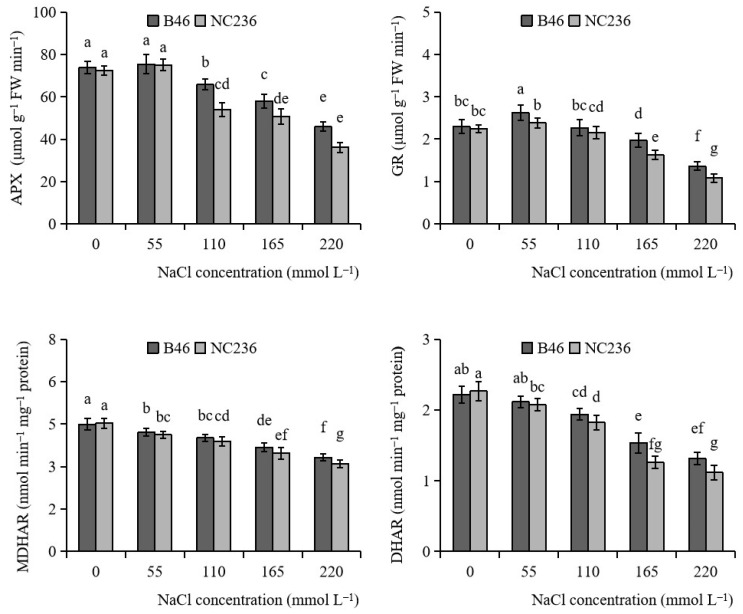
Effect of salt stress on the activities of APX, GR, MDHAR, and DHAR in maize leaves. Data are expressed as mean ± standard deviation. Different letters within the same column indicate significant difference at the 5% level.

**Figure 4 plants-11-01547-f004:**
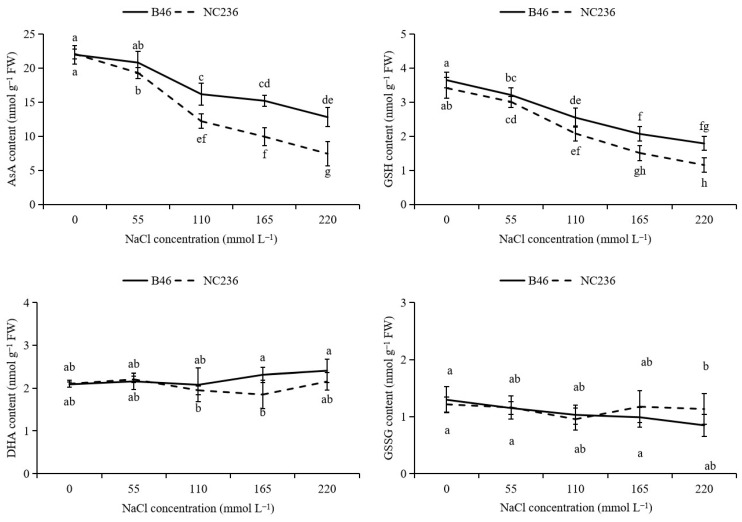
Effect of salt stress on the contents of AsA, DHA, reduced GSH, oxidized GSSG, the ratio of AsA/DHA, and GSH/GSSG in maize leaves. Data are expressed as mean ± standard deviation. Different letters within the same column indicate significant difference at the 5% level.

**Figure 5 plants-11-01547-f005:**
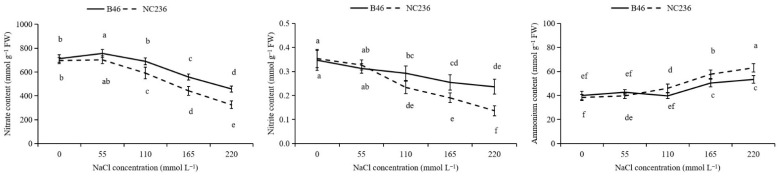
Effects of salt stress on the contents of nitrate, nitrite, and ammonium in maize leaves. Data are expressed as mean ± standard deviation. Different letters within the same column indicate significant difference at the 5% level.

**Figure 6 plants-11-01547-f006:**
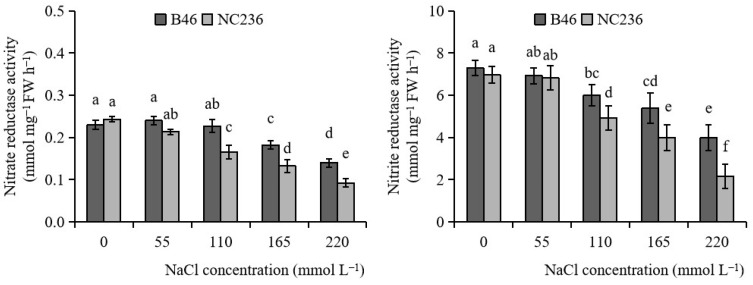
Effects of salt stress on the activities of nitrate reductase and nitrite reductase in maize leaves. Data are expressed as mean ± standard deviation. Different letters within the same column indicate significant difference at the 5% level.

**Figure 7 plants-11-01547-f007:**
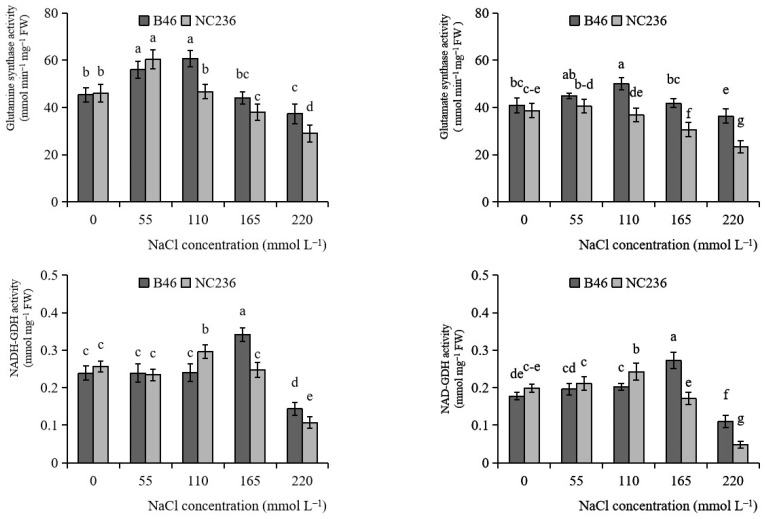
Effects of salt stress on the activities of glutamine synthase, glutamate synthase, NADH-GDH, and NAD-GDH in maize leaves. Data are expressed as mean ± standard deviation. Different letters within the same column indicate significant difference at the 5% level.

**Figure 8 plants-11-01547-f008:**
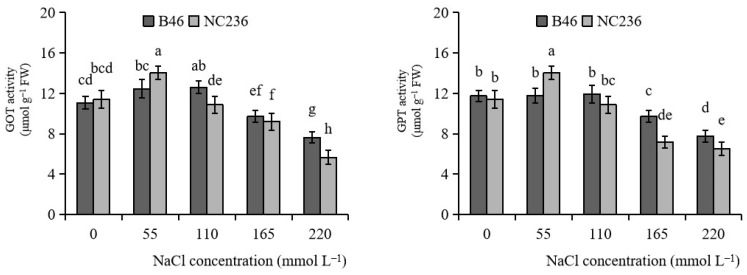
Effects of salt stress on the activities of GOT and GPT in maize leaves. Data are expressed as mean ± standard deviation. Different letters within the same column indicate significant difference at the 5% level.

**Figure 9 plants-11-01547-f009:**
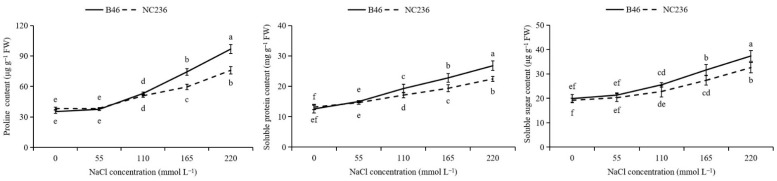
Effects of salt stress on the contents of proline, soluble protein, and soluble sugar in maize leaves. Data are expressed as mean ± standard deviation. Different letters within the same column indicate significant difference at the 5% level.

**Table 1 plants-11-01547-t001:** Effects of salt stress on the contents of Na^+^, K^+^, and K^+^/Na^+^ ratio in maize leaves.

NaCl Concentration (mmol L^−1^)	B46			NC236		
	Na^+^ Content (%)	K^+^ Content (%)	K^+^/Na^+^ Content (%)	Na^+^ Content (%)	K^+^ Content (%)	K^+^/Na^+^ Ratio (%)
0	0.74 ± 0.10 ^e^	5.48 ± 0.21 ^a^	7.44 ± 0.78 ^a^	0.78 ± 0.10 ^e^	5.49 ± 0.17 ^a^	7.08 ± 0.85 ^a^
55	1.53 ± 0.12 ^d^	5.28 ± 0.32 ^ab^	3.45 ± 0.07 ^b^	1.63 ± 0.05 ^d^	5.23 ± 0.11 ^b^	3.20 ± 0.16 ^b^
110	2.15 ± 0.26 ^c^	5.10 ± 0.05 ^bc^	2.38 ± 0.19 ^c^	2.35 ± 0.06 ^c^	5.06 ± 0.05 ^bc^	2.15 ± 0.07 ^c^
165	2.67 ± 0.15 ^b^	4.94 ± 0.09 ^cd^	1.86 ± 0.12 ^cd^	3.15 ± 0.08 ^b^	4.75 ± 0.11 ^d^	1.51 ± 0.02 ^d^
220	3.19 ± 0.16 ^a^	4.79 ± 0.13 ^d^	1.51 ± 0.10 ^d^	3.62 ± 0.11 ^a^	4.49 ± 0.13 ^e^	1.24 ± 0.01 ^d^

Note: Data are expressed as mean ± standard deviation. Different letters within the same column indicate significant difference at the 5% level.

## Data Availability

The data presented in this study are available upon request from the corresponding author.

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
