# Peer review of "The Involvement of Antioxidant Enzyme System, Nitrogen Metabolism and Osmoregulatory Substances in Alleviating Salt Stress in Inbred Maize Lines and Hormone Regulation Mechanisms"

_plants, 2022, doi:10.3390/plants11121547_

Round 1
Reviewer 1 Report
It is very relevant study in the context of climate change and abiotic stress adaptation. Authors have designed the study very well and have taken significant efforts to present their research findings. Use of different saline solutions and measuring different salinity tolerance parameters is very important to comprehend how salt tolerance mechanism works at phenotypic, biochemical, physiological and genotypic level. I think study may help peer maize researchers in addressing this issue. However, there is a weakness of this study and it is that authors have only used two genotypes to evaluate the effect of salinity under varied saline environments and both genotypes appear to be from North America (precisely from USA) and I wonder why authors did not added genotypes those are of local origin. Additionally for breeders, understanding genetic background is very important part of crop improvement and this study has left this space to a greater extent.
Also, in some places, use of ENglish is very poor as setences do not read smooth and very awaward to read. I recommend authors to review this manuscript with native English speaker to review it additionally.
I stonrgly feel that author needs to make major revision to be further considered this manuscript. For other sections, please see the attached review report.

Author Response
Dear Reviewer 1,
Thank you for your comments concerning my manuscript. Those comments are all valuable and very helpful for revising and improving manuscript. We have studied comments carefully and have made correction which we hope meet with approval. Revised portion are marked in red in the manuscript. The main corrections in the manuscript and the responds to the reviewers’ comments are as following:
Comment 1:It is a very relevant study in the context of climate change and abiotic stress adaptation. Authors have designed the study very well and have taken significant efforts to present their research findings concisely. Use of different saline solutions and measuring different salinity tolerance parameters is very important to comprehend how salt tolerance mechanism works at phenotypic, biochemical, physiological and genotypic level. I think this study may help peer maize researchers in addressing this issue.
Reply 1:Thank you very much for your recognition and appreciation of our research topics. Your encouragement is our greatest motivation. Our research team began to pay attention to the sustainable development of maize in saline alkali land in 2012. China is a country with serious soil salinization, which is widely distributed and covers a large area. There are about 100million hm2 of saline soil, and about 4/5 of the saline soil has not been effectively developed and utilized, which greatly hinders the development of agriculture in China. We identified the salt tolerance of 242 maize inbred lines through hydroponic experiments in the artificial climate room, and classified them according to the degree of salt tolerance. Based on the evaluated salt tolerant inbred lines and salt sensitive inbred lines, we carried out research from the perspective of physiology, agronomic phenotype, transcriptome and metabolism, and discussed the differences of salt tolerance mechanisms between them, so as to provide technical support and theoretical basis for cultivating salt tolerant maize varieties.
Comment 2:However, there is a weakness of this study and it is that only two genotypes have been included to evaluate the effect of salinity under varied saline environments and both genotypes appear to be from North America (precisely from USA) and I fail to understand how authors would be able to assess the genotype suitable for local environment. For breeders, understanding genetic background is very important part of crop improvement and this study has left this space to a greater extent.
Reply 2:Thank you for your modification suggestions. Indeed, as you said, this is a very important issue. Your modification suggestions are very helpful for the presentation of the manuscript in a more clear and logical way. We will reply from the following three aspects.
First, we selected 242 maize inbred lines in the preliminary experiment, including 153 self bred inbred lines, 89 introduced inbred lines, 9 TS germplasm resources, 99 SS germplasm resources, 1 Pop germplasm resources, 103 NSS germplasm resources and 30 Mixed germplasm resources. Please refer to the attachment, table number and origins of provided maize inbred line (242).
Secondly, we actually carry out the experiment according to your ideas. We have selected materials from different genetic backgrounds to better evaluate the characteristics of salt tolerance of maize inbred lines. We conducted hydroponic experiments and pot experiments, based on salt tolerance coefficient and correlation coefficient analysis, principal component analysis and comprehensive evaluation of salt tolerance. The salt tolerance of 242 maize inbred lines at seedling stage was divided into 5 categories, i.e. strong salt tolerance, including 12 maize inbred lines such as B46. Moderate salt tolerance, including 42 maize inbred lines such as NC320. General salt tolerance, including 58 maize inbred lines such as 50K-4-1. Common salt sensitive, including 77 maize inbred lines such as H0480. High salt sensitive type, including 53 maize inbred lines such as NC236. Based on the screening results, we found that B46 and nc236 had great differences in salt resistance, because we chose to carry out subsequent experiments.
Thirdly, among the numerous maize germplasm resources, we used the above resistant and sensitive maize inbred lines as test materials, and used high-throughput sequencing technology to determine the molecular mechanism of Maize Salt Tolerance under salt stress by using comparative transcriptome method. In our follow-up studies, WRKY transcription factor was identified as the most important transcription factor in the salt resistance regulatory network of maize. It is also clear that hormone biosynthesis and hormone signal transduction pathways are closely related to plant salt resistance. The results of transcriptome analysis showed that the expression level of antioxidant enzymes in salt resistant maize inbred lines was high, showing a strong ability to scavenge reactive oxygen species. We also screened the key genes of anti salt transcription factors, hormone synthesis and signal transduction, and active oxygen scavenging.
We look forward to your approval of the above reply and your further guidance.
Comment 3:Use of English is very poor as the sentences do not read smooth and very awkward in nature. I recommend authors to review the manuscript with native English speaker to review it additionally. I strongly feel that author needs to make significant major revision to further consider this manuscript.
Reply 3:We apologize for the poor language in our manuscript. We have worked on the manuscript for a long time. The repeated addition and deletion of sentences and chapters obviously lead to poor readability. We have retouched and modified the manuscript through AJE(https://www.aje.cn). At the same time, we worked on language and readability, and had native English speakers make language corrections. At present, the language of the entire manuscript has been substantially improved. We sincerely hope to receive your further guidance.
Comment 4:Abstract is too lengthy and way beyond the threshold of 200 words. I encourage authors to trim it further and make it succinct to the point.
Reply 4:Thank you for pointing out this issue. We have revised and refined to make abstract succinct to the point. Best.
Comment 5:Introduction: Authors have provided sufficient information on the background and supplemented with relevant literature.
Reply 5:Thank you for your support and recognition. In the preface, we reviewed the latest research progress at home and abroad in order to better explain the core points of our research and the key problems to be solved.
Comment 6:Lines 55-57: “The plant leaves stomata were closed, the process of photosynthetic carbon assimilation was inhibited, and the antioxidant capacity of plants was reduced under salt stress [7,8].”. This sentence should be presented differently as it reads completely in the past, needs revision.
Reply 6:Based on helpful and careful suggestions from reviewer,we have revised and improved the sentences in the manuscript to present them in a clearer way, especially deleting some wordy expressions and repeated sentences.
Comment 7:Results: Results are presented well but too many figures (total of 11 figures and 1 tables) have been presented, which makes the manuscript very redundant. I suggest that authors present relevant figures in the manuscript and move other figures to the supplementary information. Also, numbers are being repeated in the text despite they are being presented in the respective figures.
Reply 7:Thank you very much. We agree with this helpful suggestions and have modified some parts properly according to journal’s notes.
Comment 8:Discussion: Discussion sounds more like introduction as it lacks the robust discussion of presented results. Authors must improve the discussion as it lacks connecting results findings and how these results could impact further investigation. Also paragraphs in the discussion section are too long and need to be shortened based on the discussed idea.
Reply 8:We are very grateful for you raising this important question. We have refined the manuscript to better illustrate the involvement mechanisms of antioxidant enzyme system, nitrogen metabolism and osmoregulatory substances in alleviating salt stress in inbred maize lines and hormone regulation.
Comment 9:Materials and methods: Overall experiment is designed well and authors have provided enough details so that peer researchers can repeat this experiment. Only issue I feel is that authors must have used more number of accessions/varieties to assess this study so that effect of genetic background could have also been assessed robustly.
Reply 9:Thank you for your suggestion. In fact, we conducted the experiment according to your idea. The screening results of 242 varieties in the preliminary experiment were not included in the manuscript. In the future, we will strengthen the discussion in the experiment in order to better show our research results.
Comment 10:Conclusion: Authors can further improve conclusion section as more key interpretations can be added in addition to existing take home messages (THMs).
Reply 10:Your modification suggestions are very helpful for the presentation of the manuscript in a clearer and logical way. We have made modifications according to your requirements and the submission requirements of the editorial department.
Comment 11:References: Too many references (a total of 94 references) and I think some references are redundant and repetitive so can be further trimmed with presenting only relevant references.
Reply 11:Thank you for your suggestion. We have deleted some references and reduced the number of references from 94 to 85, so as to more clearly express our research results and previous research contributions.
Special thanks to you for your instructive comments.` Our research team sincerely looks forward to your approval of the above reply. If you have any other questions, please feel free to contact us.
Sincerely yours,
Mingquan Wang, Shichen Gong, Lixin Fu, Guanghui Hu*, Guoliang Li, Shaoxin Hu and Jianfei Yang
Maize Research Institute, Heilongjiang Academy of Agricultural Sciences, China
5 June, 2022

Reviewer 2 Report
The research presented in the manuscript entitled "The Involvement of Antioxidant Enzyme System, Nitrogen Metabolism and Osmoregulatory Substances in Alleviating Salt Stress in Inbred maize Lines and Hormone Regulation Mechanisms” by Mingquan Wang, Shichen Gong, Lixin Fu, Guanghui Hu, Guoliang Li, Shaoxin Hu and Jianfei Yang concerns salt stress. Salt stress affects crop growth and yield. As the object of the study, the authors chose one of the most important plant species involved in the production of seeds that form the basis of human nutrition. This plant is Zea mays and in their research the authors used two inbred maize lines. One from them is a salt-tolerant inbred line B46 and the second is a conversely salt-sensitive inbred line marked as NC236. The authors used four different salt concentrations as stress factors. The authors analyzed the changes in antioxidant enzyme system, nitrogen metabolism, ion content, osmoregulation substances, and hormone content. These analyzes were carried out in both of the maize lines subjected to salt stress.
In the Introduction, the authors underlined the importance of salt stress as a major environmental factor that adversely affects plant growth, development, and agricultural production. The authors give some information from the available literature and pay particular attention to the mechanisms of various physiological processes.
In the second part of the publication, entitled 2. Results, the authors present the results of eleven experiments. They are described in a very clear manner, with detailed scatter plot and bar graph. Their analysis shows that they used various biochemical techniques and techniques used in plant physiology. These methods are selected and used properly to answer the research questions.
In part 3. Discussion of the paper, the results obtained by the authors are thoroughly analyzed and discussed using the data obtained. The results obtained by the authors are compared to the extensive literature data.
The results obtained by the authors allowed for the formulation of conclusions that were collected in the final part of the manuscript.
Minor note:
In the “Results” part of this paper each subsection starts from “As shown in Figure …”. It may be worthwhile to diversify the beginning of the descriptions, which will make this part of the publication more interesting for readers.
The experiments presented in this work are correctly and properly carried out and discussed in detail. There is no doubt that the reviewed paper will be interesting for readers, which will translate into its citation rate. I would not like to go into details about the work because it will repeat the content contained therein. To sum up, I find this paper very interesting and I think that the reviewed paper is suitable for publishing in “Plants”.
Author Response
Dear Reviewer 2,
Thank you for your comments concerning my manuscript. Those comments are all valuable and very helpful for revising and improving manuscript. We have studied comments carefully and have made correction which we hope meet with approval. Revised portion are marked in red in the manuscript. The main corrections in the manuscript and the responds to the reviewers’ comments are as following:
Comment 1:The research presented in the manuscript entitled "The Involvement of Antioxidant Enzyme System, Nitrogen Metabolism and Osmoregulatory Substances in Alleviating Salt Stress in Inbred maize Lines and Hormone Regulation Mechanisms” by Mingquan Wang, Shichen Gong, Lixin Fu, Guanghui Hu, Guoliang Li, Shaoxin Hu and Jianfei Yang concerns salt stress. Salt stress affects crop growth and yield. As the object of the study, the authors chose one of the most important plant species involved in the production of seeds that form the basis of human nutrition. This plant is Zea mays and in their research the authors used two inbred maize lines. One from them is a salt-tolerant inbred line B46 and the second is a conversely salt-sensitive inbred line marked as NC236. The authors used four different salt concentrations as stress factors. The authors analyzed the changes in antioxidant enzyme system, nitrogen metabolism, ion content, osmoregulation substances, and hormone content. These analyzes were carried out in both of the maize lines subjected to salt stress.
Reply 1:Thank you for your appreciation and recognition of our scientific research topics. Your encouragement is the biggest driving force for our scientific research. In the future, our research team will explore the molecular mechanism of salt stress tolerance in maize from the perspective of transcriptome and metabolomics.
Comment 2:In the Introduction, the authors underlined the importance of salt stress as a major environmental factor that adversely affects plant growth, development, and agricultural production. The authors give some information from the available literature and pay particular attention to the mechanisms of various physiological processes. In the second part of the publication, entitled 2. Results, the authors present the results of eleven experiments. They are described in a very clear manner, with detailed scatter plot and bar graph. Their analysis shows that they used various biochemical techniques and techniques used in plant physiology. These methods are selected and used properly to answer the research questions.
Reply 2:Thank you for your support to our scientific research work and look forward to your further guidance. Our research team began to pay attention to the sustainable development of maize in saline alkali land in 2012. China is a country with serious soil salinization, which is widely distributed and covers a large area. There are about 100million hm2 of saline soil, and about 4/5 of the saline soil has not been effectively developed and utilized, which greatly hinders the development of agriculture in China. We identified the salt tolerance of 242 maize inbred lines through hydroponic experiments in the artificial climate room, and classified them according to the degree of salt tolerance. Based on the evaluated salt tolerant inbred lines and salt sensitive inbred lines, we carried out research from the perspective of physiology, agronomic phenotype, transcriptome and metabolism, and discussed the differences of salt tolerance mechanisms between them, so as to provide technical support and theoretical basis for cultivating salt tolerant maize varieties.
Comment 3:In part 3. Discussion of the paper, the results obtained by the authors are thoroughly analyzed and discussed using the data obtained. The results obtained by the authors are compared to the extensive literature data.The results obtained by the authors allowed for the formulation of conclusions that were collected in the final part of the manuscript.
Reply 3:Thank you for your approval. Our research team will continue to strive for more research results in the physiological and ecological mechanisms of salt stress tolerance and its regulation in maize.
Comment 4:Minor note: In the “Results” part of this paper each subsection starts from “As shown in Figure …”. It may be worthwhile to diversify the beginning of the descriptions, which will make this part of the publication more interesting for readers.
Reply 4:Your suggestions are very helpful for the presentation of the manuscript in a clearer and logical way. We have made modifications according to your requirements and the submission requirements of the editorial department. Thank you very much.
Comment 5:The experiments presented in this work are correctly and properly carried out and discussed in detail. There is no doubt that the reviewed paper will be interesting for readers, which will translate into its citation rate. I would not like to go into details about the work because it will repeat the content contained therein. To sum up, I find this paper very interesting and I think that the reviewed paper is suitable for publishing in “Plants”.
Reply 5:Thank you for your appreciation and recognition of our scientific research topics. Best.
Special thanks for your comments. In addition, we have revised and improved the preface, language grammar and refined the references to meet the “Plants” standard. Our research team sincerely looks forward to your approval of the above reply.
Sincerely yours,
Mingquan Wang, Shichen Gong, Lixin Fu, Guanghui Hu*, Guoliang Li, Shaoxin Hu and Jianfei Yang
Maize Research Institute, Heilongjiang Academy of Agricultural Sciences, China
5 June, 2022
Round 2
Reviewer 1 Report
Authors have made the corrections and manuscript has been improved significantly. However, I am still not convince that the manuscript can be accepted to be publish in the present form. Please see the below specifc comments:
1. Abstract: Revised abstract has 464 words and by jounal format the abstract should be <200 words so not sure how authors adhered with the journal guidelines.
2. Number of Figures: There are total of 11 figures and authors indicated that they have made some corrections according to the journal format in the revised draft. I do not see any reflected changes that would address suggested review. I still feel that the manuscript reads redundant at places and some figures should be moved to supplementary section.
3. Concerning use of 242 varieties/accessions, I would suggest that authors indicate this in the materials and methods, result, and discussion sections so that it would help readers to comprehend the rational of using only two genotypes for this study.
Author Response
- Abstract: Revised abstract has 464 words and by jounal format the abstract should be <200 words so not sure how authors adhered with the journal guidelines.
Reply: The abstract should be a total of about 200 words maximum. We are very sorry that we failed to refine the summary for the first time. According to your suggestion, we have refined the abstract to less than 200 words.
- Number of Figures: There are total of 11 figures and authors indicated that they have made some corrections according to the journal format in the revised draft. I do not see any reflected changes that would address suggested review. I still feel that the manuscript reads redundant at places and some figures should be moved to supplementary section.
Reply: Thank you for your pertinent suggestions on the layout format. We have changed Figure 10 and Figure 11 into supplementary documents, figure S1 and figure S2 respectively.
- Concerning use of 242 varieties/accessions, I would suggest that authors indicate this in the materials and methods, result, and discussion sections so that it would help readers to comprehend the rational of using only two genotypes for this study.
Reply: Thank you for your valuable suggestion, which is helpful for readers to understand the contents of the manuscript. We have added the description of this part in the materials and methods of the manuscript. The details are as follows. “Based on the results of preliminary experiments, we selected 242 maize inbred lines in the preliminary experiment, including 9 TS germplasm resources, 99 SS Germplasm resources, 1 POP germplasm resources, 103 NSS germplasm resources and 30 MIXED germplasm resources. We conducted hydroponic experiments and pot experiments, based on salt tolerance coefficient and correlation coefficient analysis, principal component analysis and comprehensive evaluation of salt tolerance. Based on the screening results, we found that B46 and NC236 had great differences in salt resistance. Therefore, B46 and NC236 were selected as the experimental material to better explain the difference characteristics of salt resistance.” Special thanks to you for your instructive comments. Our research team sincerely looks forward to your approval of the above reply.
Round 3
Reviewer 1 Report
Authors made the suggested correction so I do not have furher comments on this manuscript except
Line 341: Listed "Table S1" should be "Table 1"